# Environmental Impact of the Average Hong Kong Diet: A Case for Adopting Sustainable Diets in Urban Centers

**Tsz Wing Tang and Tanja Sobko \*** 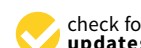

School of Biological Sciences, Faculty of Science, The University of Hong Kong, Hong Kong, China; ltanghk@connect.hku.hk

\* Correspondence: tsobko@hku.hk; Tel.: +852-2299-0611

**Abstract:** As global population growth continues, the rapidly increasing demand for food and the environmental impact of this demand is a growing concern. Most food in Hong Kong is imported, which has implications for the associated environmental footprint. The carbon and water footprints (CF and WF) of the average Hong Kong diet were estimated from available sources and compared to well-accepted sustainable diets to characterize environmental sustainability. The total CF was 5701.90 g $CO_2$-eq per capita/day, and the WF was 4782.31 L per capita/day. While meat products contributed only 22% to the weight, they were responsible for 57% and 53% of the total CF and WF, respectively. The impacts of the Hong Kong diet were greater than those of well-accepted sustainable diets, possibly due to the heavy consumption of meat and the import of foods. This confirms an urgency to increase environmental awareness among Hong Kong's consumers and make interventions toward the adoption of sustainable, plant-based diets.

**Keywords:** carbon footprint; water footprint; meat consumption; diet sustainability

## 1. Introduction

The demand for food is steadily increasing with the rise in global population. Globally, the food system is one of the most economically important sectors, making up a huge portion of the gross domestic product (GDP) of most countries. As such, the environmental impact of the food system is a growing concern worldwide [1]. A three-fold increase in Chinese meat consumption has been observed over the past five decades, which raises particularly alarming concerns for related greenhouse gas emissions, as meat-based diets are associated with at least four times more emissions than plant-based diets [2]. The significance of dietary habits for the global environment is reflected in the 2030 Sustainable Development Goals, as several objectives are closely linked to the production, use, and disposal of food [3].

At every phase in the food system, including producing, processing, packaging, transporting, storing, and retailing, there is an environmental cost that is closely associated with climate change and environmental sustainability [4,5]. As a result of an imbalance in the entire system, the environment is being degraded—deforestation for the creation of farmland, in turn, reduces biodiversity and intensifies water scarcity [6]. Here, we chose to analyze the environmental impact of post-production consumption, which is estimated to contribute to more than half of total food-related GHG emissions [5]. The distance food is transported has a direct correlation to its GHG emissions [5], which is particularly relevant to highly urbanized locations, such as Hong Kong [7]: since local food production in Hong Kong is minimal, consumers are heavily reliant on imported foods.

Environmental impact can be quantified by several parameters, such as energy consumption per calorie of food, area of land use and area of land degraded, amount of waste generated, and the

amount of chemical fertilizers and pesticides applied—all of which allow for useful comparisons and future environmental predictions [5,8–10]. The most common indicators of food-related environmental impacts by the food system are the carbon footprint (CF), the water footprint (WF), and the ecological footprint (EF) [11].

The CF of the food system refers to the quantity of direct and indirect emissions of greenhouse gasses (GHG) accumulated throughout the life stage of a food product [11]. Direct emissions include nitrogen oxides and methane from fertilizer use and ruminant rearing. Indirect emissions include the effects of replacing forests with croplands and pasturelands [5]. The total amount of greenhouse gas emissions is quantified by the mass of carbon dioxide equivalent emitted (kg $CO_2$-eq) [11].

WF (measured by the volume of freshwater used per unit of time) is a consumption-based measurement of total freshwater used throughout the life cycle of food products [11]. Similar to that of CF, the calculation of WF also considers both direct and indirect water use, including the use of surface and groundwater (blue WF) together with the consumption of soil-locked rainwater (green WF) [11].

Lastly, EF is a measurement of combined environmental impacts of human activities on the world's ecosystem by accounting for the use of resources and carbon dioxide emission [11], such as the impact of conversion of rainforest acreage into a coffee plantation. To compute EF, six ecosystem services associated with six particular types of land use are assessed: (1) cropland, (2) grazing land, (3) fishing ground, (4) forest, (5) carbon sink (carbon uptake land), and (6) build-up area [11]. EF, usually expressed in the unit of area of land use in terms of the world average of bio-productivity, is much more complex to estimate, and therefore most studies reported CF and WF as measurements of environmental impact [7,12–16]. Our study also follows this norm and primarily takes CF and WF into account.

The modern food system is unable to sustain a growing population in an environmentally-friendly manner [4]. The food system's contribution to global anthropogenic greenhouse gas emissions in 2008 was 19%–29%, which is equivalent to 9800–16,900 megatonnes of carbon dioxide emitted [5]. By 2050, the food production process will contribute to an 80% increase in greenhouse gas emissions and lead to more land clearing for agriculture use [4]. With climate change and GHG emissions already out of historical precedent, such an increase is hardly sustainable. To ensure quality of life and meet population needs, sustainable food systems will need to be evaluated and implemented. In order to achieve a sustainable food system, three remedial approaches have been suggested: (1) to increase production efficiency, (2) to lower consumption, and (3) to re-arrange the balance of power within the system [10].

Sustainable diets are defined as those that both meet nutritional guidelines while simultaneously minimizing environmental impacts [17]. Some studies suggest that a reduction in meat consumption would further shift the diet to a more sustainable one. For example, diets such as the Mediterranean diet and the New Nordic diet are generally considered to be sustainable [18], as they meet most of the above criteria. The trend of urbanization, while indicative of many of the problems in the food system, is increasingly synonymous with demand for better sustainability and environmentally-safe food. In developed Chinese cities, rising demands for food safety and environmental friendliness reflect increased awareness of food-related environmental impacts [19]. This is reflected in a current trend, especially among the younger, urban generation, to switch from traditional, omnivorous diets to vegetarianism [19].

A diet is considered sustainable when the environmental cost (i.e., the CF or WF) of the food system contributes no more than 50% of the annual global per capita GHG emissions ([20] pp. 102–105). This translates to under 750 kg $CO_2$-eq per year, or 2055 g $CO_2$-eq per capita per day ([21] p.160). However, around 52% of the global population adopts dietary habits that produce a CF higher than suggested [1]. It is, therefore, vital to promote sustainable diets in an effort to reduce GHG emissions; consequently, many groups have been developing methodologies for sustainable diet assessment [4,22–24]. Despite progress in this area, most studies focus on a single environmental indicator [1], rarely accounting for other environmental impacts [25], and only a few describe Asian dietary habits [7,13]. In the case of

Hong Kong, the most recent reports on GHG emissions only measure water consumption [26] and the average diet [7]. Thus, the need to systematically analyze and report the environmental impacts of a typical Asian or Hong Kong local dietary habit using multiple indicators is warranted. By doing so, this study hopes to highlight the problems associated with current consumers' dietary habits and advocate for a change that will positively impact the environment.

The objective of this study is to characterize the environmental sustainability of a mean Hong Kong diet by comparing its calculated impact to three well-accepted sustainable diets: (1) the vegan diet, (2) the Mediterranean diet, and (3) the New Nordic diet [27,28]. To our knowledge, no comparable studies on the WF for sustainable diets or a recommended WF value for a food system are currently available. The estimated CF of the Hong Kong diet was also compared to a recommended sustainable value of 2055 g $CO_2$-eq per capita per day [21].

## 2. Results

Each Hong Kong adult citizen consumes a total of 1845.98 g of food per day. The CF of an average diet was estimated to be 5701.90 g $CO_2$-eq per capita per day while the estimated total WF was 4782.31 L per capita per day (Table 1).

**Table 1.** Carbon footprint (CF) and water footprint (WF) of the average daily Hong Kong diet/person (ranked by weight).

| Food Item | Food Supply (g) | Food Consumed (g) | CF Index g $CO_2$-eq /g | WF Index L/g | CF g $CO_2$-eq/ cap/day | WF L/ cap/day |
|---|---|---|---|---|---|---|
| Grains | | | | | | |
| Wheat and products | 145.59 | 122.29 | 0.94 | 1.87 | 114.67 | 228.53 |
| Rice (milled equivalent) | 119.12 | 100.06 | 2.51 | 1.99 | 250.54 | 198.70 |
| Maize and products | 12.25 | 10.29 | 0.66 | 1.28 | 6.77 | 13.12 |
| Oats | 6.93 | 5.82 | 1.33 | 2.09 | 7.72 | 12.12 |
| Cereals, other | 6.49 | 5.45 | 1.33 | 1.46 | 7.24 | 7.94 |
| Barley and products | 0.11 | 0.09 | 1.33 | 1.54 | 0.12 | 0.14 |
| Vegetables | | | | | | |
| Vegetables, other | 292.93 | 246.06 | 0.93 | 0.24 | 228.27 | 58.17 |
| Potatoes and products | 67.70 | 56.87 | 0.18 | 0.89 | 10.21 | 50.55 |
| Nuts and products | 34.68 | 29.14 | 12.10 | 8.38 | 351.67 | 243.64 |
| Tomatoes and products | 8.55 | 7.18 | 0.93 | 0.82 | 6.66 | 5.87 |
| Soy beans | 7.64 | 6.42 | 1.00 | 2.22 | 6.40 | 14.25 |
| Onions | 5.86 | 4.92 | 0.93 | 0.25 | 4.57 | 1.23 |
| Beans | 3.23 | 2.72 | 1.00 | 0.37 | 2.71 | 1.01 |
| Groundnuts (shelled equivalent) | 3.15 | 2.65 | 1.33 | 8.38 | 3.52 | 22.13 |
| Cassava and products | 2.88 | 2.42 | 0.18 | 1.78 | 0.43 | 4.29 |
| Roots, other | 0.71 | 0.60 | 0.18 | 0.34 | 0.11 | 0.20 |
| Pulses, other and products | 0.36 | 0.30 | 1.00 | 3.32 | 0.30 | 0.99 |
| Pimento | 0.27 | 0.23 | 0.67 | 0.24 | 0.15 | 0.05 |
| Sweet potatoes | 0.25 | 0.21 | 0.93 | 0.33 | 0.19 | 0.07 |
| Peas | 0.05 | 0.05 | 1.00 | 3.36 | 0.05 | 0.15 |
| Sesame seed | 0.22 | 0.18 | No data | 8.97 | No data | 1.65 |
| Sugar cane | 0.03 | 0.02 | 0.93 | 0.20 | 0.02 | 0.00 |
| Sunflower seed | 0.03 | 0.02 | No data | 3.17 | No data | 0.07 |
| Fruits | | | | | | |
| Fruits, other | 96.08 | 80.71 | 0.67 | 0.87 | 53.94 | 70.29 |
| Oranges, mandarins | 50.55 | 42.46 | 0.67 | 0.68 | 28.38 | 28.75 |
| Apples and products | 22.93 | 19.26 | 0.67 | 2.48 | 12.87 | 47.70 |
| Bananas | 19.64 | 16.50 | 0.67 | 0.76 | 11.03 | 12.46 |
| Grapes and products (exclude wine) | 10.03 | 8.42 | 0.67 | 1.07 | 5.63 | 9.00 |
| Grapefruit and products | 6.99 | 5.87 | 0.67 | 0.45 | 3.92 | 2.65 |
| Lemons, limes and products | 6.47 | 5.43 | 0.67 | 0.58 | 3.63 | 3.16 |
| Pineapples and products | 4.19 | 3.52 | 0.67 | 0.67 | 2.35 | 2.36 |
| Coconuts | 2.88 | 2.42 | 0.67 | 2.29 | 1.62 | 5.52 |
| Citrus, other | 0.88 | 0.74 | 0.67 | 0.87 | 0.49 | 0.64 |
| Dates | 0.55 | 0.46 | 0.67 | 2.18 | 0.31 | 1.00 |
| Olives (including preserved) | 0.22 | 0.18 | 0.67 | 2.24 | 0.12 | 0.41 |

**Table 1.** *Cont.*

| Food Item | Food Supply (g) | Food Consumed (g) | CF Index g $CO_2$-eq /g | WF Index L/g | CF g $CO_2$-eq/ cap/day | WF L/ cap/day |
|---|---|---|---|---|---|---|
| Meat | | | | | | |
| Pig meat | 183.86 | 154.44 | 4.19 | 5.37 | 645.53 | 826.71 |
| Poultry meat | 150.30 | 126.25 | 3.41 | 3.86 | 429.46 | 485.88 |
| Offals, edible | 73.34 | 61.61 | 12.10 | 4.82 | 743.62 | 296.28 |
| Bovine meat | 70.74 | 59.42 | 21.36 | 14.96 | 1266.11 | 886.99 |
| Meat, other | 8.52 | 7.16 | 12.10 | 6.30 | 86.39 | 44.97 |
| Mutton and goat meat | 6.08 | 5.11 | 10.44 | 8.71 | 53.21 | 44.39 |
| Aquatic animals | | | | | | |
| Marine fish, other | 61.21 | 51.41 | 3.85 | 1.81 | 197.45 | 92.72 |
| Mollusks, other | 34.22 | 28.74 | 3.85 | 0.61 | 110.39 | 17.38 |
| Crustaceans | 32.47 | 27.27 | 3.85 | 0.61 | 104.74 | 16.49 |
| Freshwater fish | 27.70 | 23.27 | 3.85 | 7.01 | 89.36 | 162.65 |
| Demersal fish | 20.33 | 17.08 | 3.85 | 1.81 | 65.58 | 30.80 |
| Cephalopods | 8.36 | 7.02 | 3.85 | 0.61 | 26.96 | 4.24 |
| Pelagic fish | 7.04 | 5.91 | 3.85 | 1.81 | 22.71 | 10.67 |
| Aquatic animals, others | 0.93 | 0.78 | 3.85 | 3.14 | 3.01 | 2.45 |
| Egg | | | | | | |
| Eggs | 39.73 | 33.37 | 3.23 | 2.84 | 107.52 | 94.40 |
| Milk | | | | | | |
| Milk—Excluding butter | 290.00 | 243.60 | 1.43 | 0.95 | 347.49 | 230.61 |
| Cream | 0.52 | 0.44 | 5.92 | 1.22 | 2.58 | 0.53 |
| Fat/Oil, Salt, and Sugar | | | | | | |
| Sugar (raw equivalent) | 89.97 | 75.58 | 0.52 | 1.40 | 39.20 | 105.66 |
| Soy bean oil | 15.48 | 13.00 | 2.97 | 4.12 | 38.52 | 53.40 |
| Fats, animals, raw | 13.89 | 11.67 | 7.34 | 4.06 | 85.43 | 47.20 |
| Sweeteners, other | 7.01 | 5.89 | 2.33 | 0.18 | 13.69 | 1.07 |
| Cocoa beans and products | 4.27 | 3.59 | No data | 21.98 | No data | No data |
| Butter, ghee | 3.40 | 2.85 | 7.34 | 5.16 | 20.89 | 14.69 |
| Rape and mustard oil | 3.29 | 2.76 | 2.97 | 3.07 | 8.18 | 8.45 |
| Oil crops oil, other | 2.38 | 2.00 | 2.97 | 2.24 | 5.93 | 4.48 |
| Groundnut oil | 2.25 | 1.89 | 2.97 | 3.71 | 5.59 | 6.98 |
| Maize germ oil | 2.14 | 1.80 | 2.97 | 2.17 | 5.32 | 3.88 |
| Olive oil | 1.53 | 1.29 | 2.97 | 14.36 | 3.82 | 18.46 |
| Honey | 1.12 | 0.94 | 1.65 | No data | 1.55 | No data |
| Spices, other | 0.99 | 0.83 | 0.93 | 6.62 | 0.77 | 5.47 |
| Sunflower seed oil | 0.52 | 0.44 | 2.97 | 6.39 | 1.30 | 2.79 |
| Sesame seed oil | 0.52 | 0.44 | 2.97 | 20.86 | 1.30 | 9.10 |
| Coconut oil | 0.41 | 0.35 | 2.97 | 4.46 | 1.02 | 1.54 |
| Pepper | 0.16 | 0.14 | 0.93 | 7.00 | 0.13 | 0.96 |
| Oil crops, other | 0.03 | 0.02 | 2.97 | 2.24 | 0.07 | 0.05 |
| Beverages | | | | | | |
| Beer | 58.14 | 48.84 | 0.46 | 0.27 | 22.41 | 13.15 |
| Wine | 11.62 | 9.76 | 1.13 | 0.75 | 11.00 | 7.25 |
| Coffee and products | 10.63 | 8.93 | 0.37 | 16.83 | 3.30 | 149.90 |
| Beverages, alcoholic | 5.32 | 4.46 | 0.80 | 0.51 | 3.54 | 2.26 |
| Tea (including mate) | 4.03 | 3.38 | 0.06 | 8.13 | 0.20 | 27.44 |
| Beverages, fermented | 2.71 | 2.28 | No data | 0.51 | No data | 1.15 |
| Total: | 2197.59 | 1845.97 | | | 5701.90 | 4782.31 |

Among all the foods consumed, meat (excluding aquatic animal consumption) contributes up to 22% of the average Hong Kong diet. Animal-derived products (such as eggs, milk, and aquatic animals) contribute 24% of total food consumption, while vegetables, grains, and fruit contribute another 43% of total food consumption. The rest of the diet is composed of fat/oil, salt, sugar, beverages, and eggs (Table 2). Despite constituting only 22% of the weight of the average daily diet, the meat category contributed 57% of total CF and 53% of total WF of daily food consumption. In contrast, vegetables contributed only 11% of total CF and 8% of total WF, while comprising only 20% of the total diet.

**Table 2.** Percentage contribution of food categories to the total food consumed, CF and WF.

| Category | Food Consumed (g) | % | CF g $CO_2$-eq/cap/day | % | WF L/cap/day | % |
|---|---|---|---|---|---|---|
| Grains | 244.01 | 13% | 387.07 | 7% | 460.54 | 10% |
| Vegetables | 359.98 | 20% | 615.27 | 11% | 404.35 | 8% |
| Fruits | 185.98 | 10% | 124.30 | 2% | 183.93 | 4% |
| Meat | 413.99 | 22% | 3224.31 | 57% | 2585.22 | 54% |
| Aquatic animals | 161.49 | 9% | 620.19 | 11% | 337.40 | 7% |
| Egg | 33.37 | 2% | 107.52 | 2% | 94.40 | 2% |
| Milk | 244.04 | 13% | 350.07 | 6% | 231.14 | 5% |
| Fat/Oil, Salt and Sugar | 125.47 | 7% | 232.72 | 4% | 284.18 | 6% |
| Beverages | 77.65 | 4% | 40.45 | 1% | 201.15 | 4% |
| Total: | 1845.98 | 100% | 5701.90 | 100% | 4782.31 | 100% |

## 3. Discussion

The environmental impacts of the average Hong Kong diet, as assessed by their CF and WF, were found to be 5701.90 g $CO_2$-eq and 4782.31 L per capita/day, respectively. The average intake of food for a single Hong Kong person was summed up to a total of 2760 kcal from the 1845 g of food consumed each day, which is higher than that of the FAO recommendation of 2500 kcal per day [3] and indicates a similar energy density when compared to the total world average food supply of 1909 g (2884 kcal/cap per day) per capita per day [3].

The CF of the average daily diet in Hong Kong is among the highest compared to other regions (Figure 1), except for the UK [29], and is almost double that of the average Chinese diet despite the geographic proximity. This might be due to the fact that dietary patterns of urban areas, such as Hong Kong, differ greatly from those of rural areas [7]. Urban areas are reported to have higher food consumption, in general, in addition to higher consumption of meat and refined food products, which intensifies their environmental burden [30].

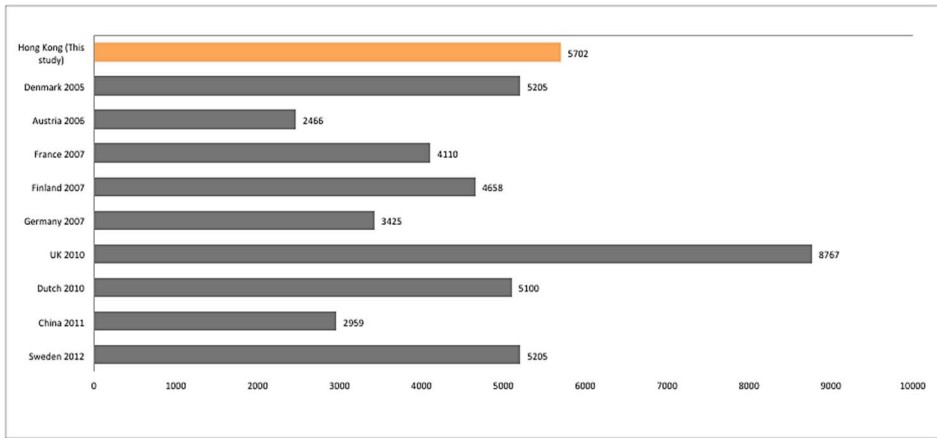

**Figure 1.** Comparison between the CF of the average Hong Kong diet and that of other regions.

The total WF for an average Hong Kong diet in this study is 4782 L per capita/day, which was similar to the 2013 Hong Kong data [7]. Hong Kong, as reported in both this and the 2013 study, tops the list when compared to other regions globally (Figure 2).

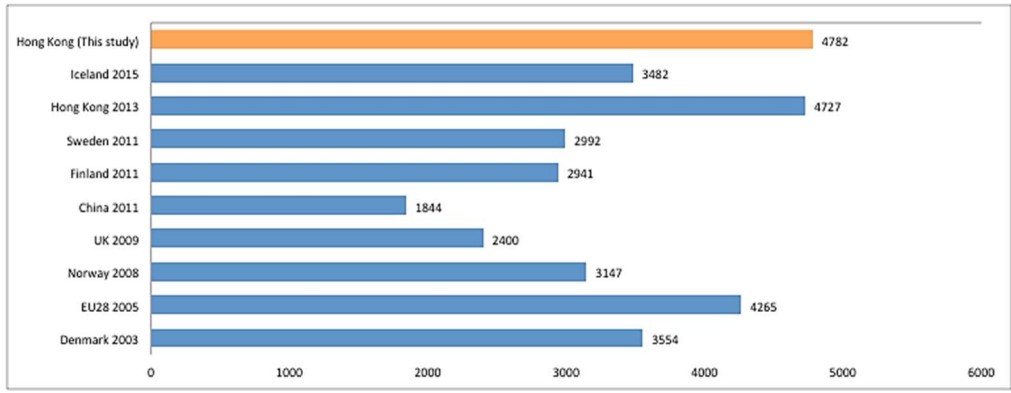

**Figure 2.** Comparison between the WF of the average Hong Kong diet and that of other regions.

The recommended values for a sustainable diet are under 2055 g $CO_2$-eq per capita/day [21], and three sustainable diets are provided as examples [25,27,28] (Figure 3). A typical Mediterranean diet often implies a low intake of red meat or processed food and a high intake of vegetables, fruit, fish, and grains [18]. The New Nordic diet is based on local food produced in the Nordic area and is characterized by a large proportion of calories provided by local non-animal products and wild foraged foods such as mushrooms, nuts, and berries [28]. The CF of the average Hong Kong diet exceeds the recommended CF value by a large extent, almost doubling the CF inherent in any of the three sustainable diets. No comparison between the WF of the average Hong Kong diet and any other sustainable diet could be done due to the lack of reliable data and water use recommendations. On evaluation, the primary cause of this massive CF is due to the high consumption of meat in Hong Kong, combined with carbon-intensive transportation of meat along with other foodstuffs.

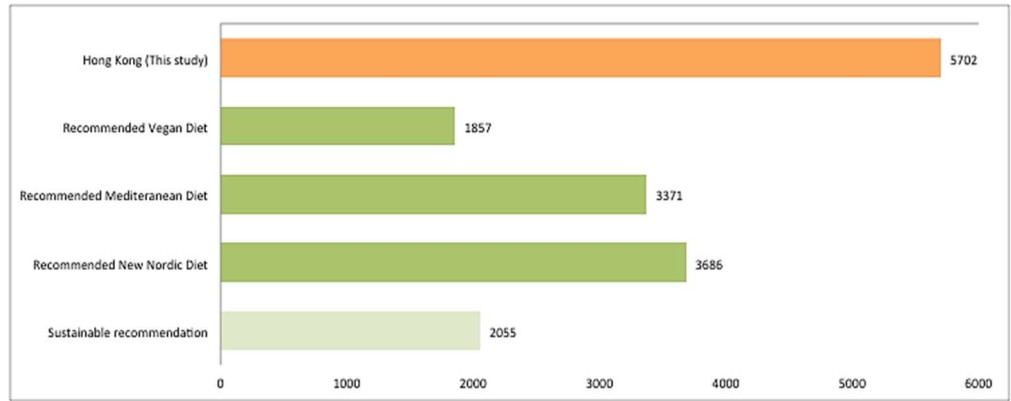

**Figure 3.** Comparison between the CF of the average Hong Kong diet and other sustainable diets.

The Hong Kong diet was found to be a typical, high-meat urban diet [31] and has one of the highest per capita meat supply and consumption rates in the world [7]. The total consumed weight of the meat category (aquatic animals, eggs, and milk excluded) was calculated to be 414 g per day per person, and accounted for 22% of total food intake. In comparison, the global average for meat consumption was only 125 g per capita/day at the time of this study, which was 7% of total food consumed [3]. It is established that meat products contribute to a larger environmental impact than any other food item in terms of GHG emissions [32,33], land use [33], and water consumption [34]. The meat consumption calculated in this paper is higher than reported by Vanham et al. [35] (361.1 g). Our results may also be an underestimation, as Yau et al. [26] recently suggested that the daily meat consumption in Hong Kong could be as high as 500 g per capita per day with a GHG contribution of 5 Mt $CO_2$-eq per year. The 22% contribution of meat on the environmental impact in our study is

57% of the total CF and 53% of the total WF of the whole diet. Meanwhile, the environmental impact of vegetable consumption is much smaller, comprising 20% of the diet and contributing only 11% to the total CF and 8% to the total WF. High intakes of meat in the average Hong Kong diet together with large amounts of general food intake, is seen to generate the diet's most negative environmental impact (Figure 4).

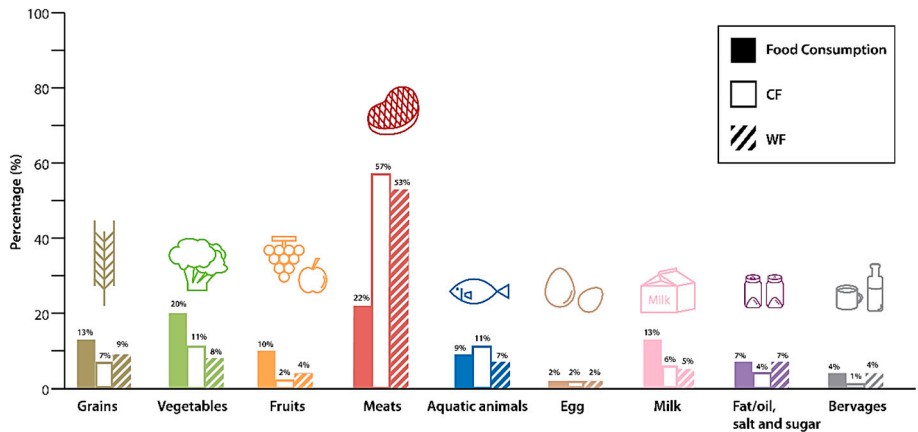

**Figure 4.** Percentage contribution of food categories to the total food consumed, CF and WF.

Meat consumption is identified in this study as the largest contributor to the unsustainable CF and WF of the total diet, while the consumption of plant-based diets is demonstrated to be sustainable. Reduced intake of protein might be associated with plant-based diets; however, the reduced intake of protein and fat that is associated with plant-based diets is generally not a problem among more economically developed countries [21], i.e., Hong Kong. In contrast, overweight, obesity, and related diseases, which are brought by overconsumption of foods that are high in added sugar, salt, and saturated fat, are more of a health concern. On the other hand, there might be a potential consumption increase when replacing meat with a plant-based diet to meet the same nutritional needs, which in turn, could potentially mitigate the sustainability of a diet. However, the CF and WF (environmental impacts) produced per kilogram of a vegetable item will still be much lower than the CF and WF per 1 kg of meat.

Considering the novelty of this topic, the standard methodology for assessing the environmental impact of a region's average diet has yet to be established. Difficulties were encountered when the estimated CF and WF of the Hong Kong diet in this study were compared to global dietary patterns, as most of the previously-reported results were not methodologically comparable. The fact that Hong Kong is both an independent region and an urban center provided an opportunity to isolate an urban dietary scenario from the rural scenarios present in other evaluations of regional diets. This study followed previously-established methodologies, where the average diet of Hong Kong was first identified. Then, the CF and WF indices from previous studies were extracted. The environmental impact was estimated by multiplying the amount of food consumed by these indices. However, the available data were partially fragmented and may have influenced the final results. Hong Kong is known to have a typical urban pattern of food supply, characterized as being heavily dependent on imported resources rather than local production [7]. The CF of food items produced in distant regions is higher than that of locally produced food [5], and thus the adoption of global CF indices might underestimate the actual CF of Hong Kong consumption, as they do not account for GHG emissions. A recent study estimated the average GHG emission in the life cycle of food items produced with a wide range of production methods [36]. If the GHG emission values were combined with food consumption data used in this study, it would result in a total GHG emissions increase of over 33,000 g $CO_2$-eq per capita/day—more than six times higher than the original calculation presented here. Therefore, the origin of food items and corresponding location-specific CF and WF indices should be taken into

account to improve the accuracy of the estimated environmental impact, including not just production, but also transportation and other processes in the evaluation. The comparison of these results with other studies had to be performed with care due to the lack of agreement on assessment criteria of environmental impacts, system boundaries, and appropriate units for both food consumption and its associated impact.

Research focused on footprint indices does not necessarily follow FAO classification; however, these indices had to be matched to the average diet taken from the FAO. For example, aquatic animal consumption was divided into distinct groups of marine fish, freshwater fish, demersal fish, pelagic fish, mollusks, crustaceans, cephalopods, and others. However, there was only a single CF index found for estimating for all aquatic products. Therefore, the same index was assigned to all of the above categories. The mismatching of food items for estimating footprint indices and total food consumption may, therefore, not accurately reflect the actual CF and WF. However, in this study, aquatic animals accounted for only 9% of consumption, so the effect is likely minimal.

Further, the data on other means of assessing environmental impacts, including EF, biodiversity loss, and pollution, are sparse. If available, these impacts could be included in the assessment to provide a more complete analysis of environmental impacts.

In order to better estimate the environmental impact of the average Hong Kong diet, more specific data is needed for both food consumption patterns as well as location-specific footprint indices. A population-based food consumption survey is planned to be conducted by the Hong Kong government in the near future and might provide a better estimation of the latest consumption patterns. Results from this analysis might, therefore, be useful for a more precise estimation of environmental costs associated with contemporary consumption patterns.

If Hong Kong residents follow the local government's dietary guideline for a healthy diet, meat-related GHG emissions could be reduced by 67% [26]. Despite this recommendation, the dietary patterns in China are drastically changing from plant-based to animal-based: meat consumption increased 350% from 1963 to 2003 [2]. However, the awareness of the local population of environmental issues has led to increased demand for more sustainable, vegetarian food items in China [19], and the results of the study presented here may further contribute to this trend.

## 4. Materials and Methods

### 4.1. Assessment Indicators (CF and WF)

CF was estimated as the total amount of GHG emissions of food products during their life cycle in terms of g $CO_2$-eq [11]. In addition to $CO_2$, other GHG emissions such as methane, nitrogen dioxide, hydrofluorocarbons, perfluorocarbons, and sulfur hexafluorides were also taken into account for the CF calculation. The impacts of these GHG were converted into carbon dioxide equivalent for better comparison [11]. WF is the total consumption of freshwater used throughout the life cycle of a food item, including the consumption of surface and groundwater as well as rainwater stored in the soil [11]. Total WF used in this study was calculated by the sum of blue WF (the use of surface and groundwater) and green WF (the consumption of soil-locked rainwater). This approach is consistent with recent studies investigating WFs [7,15,30,31,35,37]. The grey WF—the volume of fresh water needed to remove pollutants [11]—was not considered due to the lack of reliable data [7].

### 4.2. Estimation of the Average Hong Kong Daily Diet

To estimate the environmental impact of various dietary scenarios, the complete daily diet, rather than any single meal, was taken into account and allocated a functional unit (FU). This FU was chosen as a comparison of life cycle assessments to achieve a comparable estimate of environmental impact derived from average food consumption in Hong Kong [38].

The methodology of a regional average diet estimation follows the practice of referencing to national food consumption surveys [13,14,16,39] or the Food and Agriculture Organization of the

United Nation (FAO) food balance sheet [7] in the absence of a national food consumption survey. The daily diet scenario used in this paper was extracted from the latest reliable dataset for complete diet estimations—the FAO food balance sheet for Hong Kong (2013). It included all food supplied for human consumption in all available forms reaching households of citizens, except the loss of food items during production and post-production phases [17]. In total, 77 food items were provided, and all non-zero value food items were taken into account. The nine main groups of food items were grains, vegetables, fruits, meat, aquatic animals, eggs, milk, fat/oil, salt and sugar, and beverages (Table 1). In total, 356.2 g per capita/day was food waste, which was equivalent to 16% of the total food supply per household [40].

GHG emissions derived from food items were retrieved using previously established methods [25], and the global average values for each item were reported. The CF of beverages were retrieved from the published source [16]. However, due to the lack of reliable data on fermented beverages, cocoa beans and products, sesame seeds, and sunflower seeds, these food items were excluded (total of 0.3% of total food intake) from the final report.

The global average coefficients of food-derived WF in terms of liters of water consumed per unit weight of food items were adopted from Mekonnen and Hoekstra [34]. The WF indices for edible offal, as well as all items in the category of aquatic animals, raw animal fat, and cream, were retrieved from Vanham et al. in 2017 [7]. No reliable data were available for the WF of honey, which entailed 0.05% weight of the total food intake.

The environment impact of the average Hong Kong diet was estimated by multiplying the daily per capita consumption of the complete diet by the coefficients of CF and WF, respectively. The estimated CF and WF of mean diets were compared to those from other regions, measured by similar methodologies. Only studies with complete-diet analyses were included in the comparisons.

## 5. Conclusions

In conclusion, there is an urgent need for the development of a standardized methodology for assessing the environmental impacts of Asian diets, as 60% of the global population lives within Asia [3]. Asian consumption habits differ from Western habits [2]; hence, more studies on Asian diets are warranted in order to estimate the overall environmental impact on the global food system.

Owing to the fact that the average Hong Kong food consumption pattern is unsustainable, there is an urgent need to promote alternative dietary habits. Meat consumption is identified in this study as the largest contributor to the unsustainable CF and WF of the total diet, while the consumption of plant-based diets is demonstrated to be sustainable. Altering the standard consumption patterns characteristic of the average daily Hong Kong diet is thus critical in mitigating food-sourced environmental impacts. In the hope of shifting the average Hong Kong diet to a more sustainable pattern, there are a few ideas that one can consider, such as the development of a sustainable dietary guideline together with environmental education, as well as the investigation of new means of food production lending to a more sovereign and local food system. Considering other components of the food system, the elimination of food waste to mitigate the high CF and WF of the Hong Kong diet should also be considered as another possible solution.

**Author Contributions:** Both authors, T.W.T. & T.S. formulated the research question and designed the study. T.W.T. carried out acquisition, analysis. Both authors interpreted of the data, wrote the article and have approved this final version.

**Funding:** This research received no external funding.

**Acknowledgments:** We acknowledge the Barilla Center of Food and Nutrition of Italy for sharing the LCA study database. We thank John Doherty and Yvonne Yau for their comments. We thank Will Cheng and Kristopher Jordan for valuable comments and thorough editing.

**Conflicts of Interest:** The authors declare no conflict of interest.

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
