# Peer review of "Environmental Impact of the Average Hong Kong Diet: A Case for Adopting Sustainable Diets in Urban Centers"

_challenges, doi:10.3390/challe10020005_

Round 1

Reviewer 1 Report

Thank you for this novel and highly valuable submission in the emerging field of food sustainability. The geographic area of interest is unique in that its food supply is mainly imported. Therefore, though the average diet is a familiar variable, the environmental impacts of these choices are dependent on the management of resources in other countries. The authors explain this well in the discussion. There is definitely a need for standardized methods in measuring environmental impacts of Asian diets. This is an extremely well written, researched, and referenced article.  Line 86-88,   "A diet is considered sustainable when the environmental cost (i.e. the CF or WF) of the food system contributes no more than 50 % of the annual global per capita GHG emissions [20]." Could I please see the page number(s) for reference 20 and 21?   In line 102-108, it may be helpful to calculate the impact of animal derived products separately from plant products. So meat =22% and animal derived products = x %(eggs, milk, aquatic). This would make it easy to see the existing % of intake that is already plant based (43%).  In line 130-141, you discuss the Mediterranean and New Nordic diet but not the vegan diet.  In the conclusion, you make an important final recommendation.  "Development of a sustainable dietary guideline together with environmental education might be useful in shifting the average diet to a more sustainable pattern with lower meat intake." However, I feel it would be less reductive and more of a system's approach to include:  Investigating new means of food production lending to a more sovereign food system. Recovering/preventing food waste as a means of mitigating the high CF and WF of the HK diet. 

Author Response

Reviewer 1:

Thank you for this novel and highly valuable submission in the emerging field of food sustainability. The geographic area of interest is unique in that its food supply is mainly imported. Therefore, though the average diet is a familiar variable, the environmental impacts of these choices are dependent on the management of resources in other countries. The authors explain this well in the discussion. There is definitely a need for standardized methods in measuring environmental impacts of Asian diets. This is an extremely well written, researched, and referenced article.

Line 86-88,   "A diet is considered sustainable when the environmental cost (i.e. the CF or WF) of the food system contributes no more than 50 % of the annual global per capita GHG emissions [20]." Could I please see the page number(s) for reference 20 and 21?  

Response: Thank you for your comment. Page numbers for reference [20] has been added (p.102-105), as well as reference [21] (p.160).

The sentence has been updated as “A diet is considered sustainable when the environmental cost (i.e. the CF or WF) of the food system contributes no more than 50% of the annual global per capita GHG emissions [20] (p.102-105). This translates to under 750kg CO2-eq per year, or 2055g CO2-eq per capita per day [21] (p.160).” Lines 88-91, Introduction.

In line 102-108, it may be helpful to calculate the impact of animal derived products separately from plant products. So meat =22% and animal derived products = x %(eggs, milk, aquatic). This would make it easy to see the existing % of intake that is already plant based (43%).

Response: Thank you for your suggestion. Food consumed from animal derived products has now been separated from plant products. Meat = 22%, animal derived products (eggs, milk and aquatic animals) = 24% and plant-based products (grains, vegetables, fruits) = 43%.

The paragraph has been updated and it now reads as the following “Among all the foods consumed, meat (excluding aquatic animal consumption) contributed up to 22% of the average Hong Kong diet. Animal derived products (such as eggs, milk and aquatic animals) contributed 24% of total food consumption, while vegetables, grains and fruit contributed another 43% of total food consumption. The rest of the diet was composed of fat/oil, salt, sugar, beverages, and eggs (Table 2). Despite constituting only 22% of the weight of the average daily diet, the meat category contributed 57% of total CF and 53% of total WF of daily food consumption. In contrast, vegetables contributed only 11% of total CF and 8% of total WF, while comprising only 20% of the total diet.” Lines 113-120, Results.

In line 130-141, you discuss the Mediterranean and New Nordic diet but not the vegan diet. In the conclusion, you make an important final recommendation. "Development of a sustainable dietary guideline together with environmental education might be useful in shifting the average diet to a more sustainable pattern with lower meat intake." However, I feel it would be less reductive and more of a system's approach to include: Investigating new means of food production lending to a more sovereign food system. Recovering/preventing food waste as a means of mitigating the high CF and WF of the HK diet.

Response: Thank you for your comment and suggestion, we believe it is very valid and important to consider other aspects of the food cycle when one is suggesting possible solutions to create a sustainable diet in HK.

Thus, the final sentence of the conclusion has been rewritten “In the hope of shifting the average Hong Kong diet to a more sustainable pattern, there are a few ideas that one can consider, such as the development of a sustainable dietary guideline together with environmental education, as well as the investigation of new means of food production lending to a more sovereign and local food system. Considering other components of the food system, the elimination of food waste to mitigate the high CF and WF of the Hong Kong diet should also be considered as another possible solution.” Lines 283-288, Conclusion.

Reviewer 2 Report

The topic of the manuscript is very interesting for the journal and contributes to the knowledge of new sustainable diets. The work has a good scientific quality and makes a very complete bibliographic review on the subject and contributing new ideas.

However, reading the conclusions make you rethink this study. You affirm that:

Meat consumption is identified in this study as the largest contributor to CF and WF of the total diet The consumption of plant-based diets is demonstrated to be sustainable A more sustainable pattern with lower meat intake

I am disagree with these. I think that the authors must consider the amount of protein provided by one kiligram of plant-based diets and the amount contributed to the meat, since you have to consume more amount of vegetables to obtain the same nutritional value.

Moreover in Table 1, the results of CF and WF that you showed for meat and aquatic animals are very high. How have you accounted for these amounts?

Therefore, these questions need to be revised to be published. I recommend a major revision.

Author Response

Reviewer 2:

The topic of the manuscript is very interesting for the journal and contributes to the knowledge of new sustainable diets. The work has a good scientific quality and makes a very complete bibliographic review on the subject and contributing new ideas. However, reading the conclusions make you rethink this study. You affirm that:

Meat consumption is identified in this study as the largest contributor to CF and WF of the total diet The consumption of plant-based diets is demonstrated to be sustainable A more sustainable pattern with lower meat intake

I am disagree with these. I think that the authors must consider the amount of protein provided by one kiligram of plant-based diets and the amount contributed to the meat, since you have to consume more amount of vegetables to obtain the same nutritional value.

Response: Thank you for your comment. We did consider the reduced intake of protein that will be associated with the plant-based diets; however, we believe that the reduced intake of protein and fat that is associated with plant-based diets is generally not a problem among more economically developed countries. In contrast, overweight, obesity, and related diseases, which are brought by overconsumption of foods that are high in added sugar, salt, and saturated fat, are more of a problem, and their prevalence is increasing in Hong Kong.

We also understand that, indeed, there might be a potential consumption increase when replacing meat with a plant-based diet to meet the same nutritional needs, which in turn, would potentially mitigate the sustainability of a diet. However, the CF and WF (environmental impacts) produced per kilogram of each vegetable item will still be much lower than the CF and WF per 1 kg of meat. So in theory, comparing adding 2 kg of plant-based food and reducing 1 kg of meat in the diet: the overall net balance of CF and WF will still be reduced.

The above response has been cooperated into the discussion, “Meat consumption is identified in this study as the largest contributor to the unsustainable CF and WF of the total diet, while the consumption of plant-based diets is demonstrated to be sustainable. Reduced intake of protein might be associated with the plant-based diets; however, the reduced intake of protein and fat that is associated with plant-based diets is generally not a problem among more economically developed countries [21]. In contrast, overweight, obesity, and related diseases, which are brought by overconsumption of foods that are high in added sugar, salt, and saturated fat, are more of a health concern.

On the other hand, there might be a potential consumption increase when replacing meat with a plant-based diet to meet the same nutritional needs, which in turn, could potentially mitigate the sustainability of a diet. However, the CF and WF (environmental impacts) produced per kilogram of a vegetable item will still be much lower than the CF and WF per 1 kg of meat.” Lines 171-181. Discussion.

Moreover, as the main focus of this particular article is on environmental impacts, we believed that the statement suggested by the reviewer, referring to the nutritional impacts, would fit more in the follow-up article. We believe that this valuable discussion should be addressed separately, with a deeper literature review and further analysis. Again, the nutritional impact was not the objective of this paper.

Moreover in Table 1, the results of CF and WF that you showed for meat and aquatic animals are very high. How have you accounted for these amounts?

Response: Thank you for your comment. Yes, we have accounted for these amounts. According to Table 2, which is the summary of findings from Table 1, Meat makes up only 22% in HK diet, but contributes 57% of total carbon footprint and 54% of total water footprint. However, aquatic animals only make up 9% of total HK diet consumed and contributes 11% total CF and 7% total WF. By comparison, it can be clearly seen that meat is creating a larger impact on the environment, which is the main focus of the article and has been addressed throughout.

Specifically, the reason for high CF and WF from Hong Kong diet was accounted for and discussed in the following paragraph

“On evaluation, the primary cause of this massive CF is due to the high consumption of meat in Hong Kong, combined with carbon-intensive transportation of meat along with other food stuffs.

The Hong Kong diet was found to be a typical, high-meat urban diet [31] and has one of the highest per capita meat supply and consumption rates in the world [7]. The total consumed weight of the meat category (aquatic animals, eggs and milk excluded) was calculated to be 414 g per day per person, and accounted for 22% of total food intake. In comparison, the global average for meat consumption was only 125 g per capita/day at the time of this study, which was 7% of total food consumed [3]. It is established that meat products contribute to a larger environmental impact than any other food item in terms of GHG emissions [32, 33], land use [33], and water consumption [34]. The meat consumption calculated in this paper is higher than reported by Vanham et al. [35] (361.1 g). Our results may also be an underestimation, as Yau et al. [26] recently suggested that the daily meat consumption in Hong Kong could be as high as 500 g per capita per day with a GHG contribution of 5Mt CO2-eq per year.” Lines 151-164, Discussion.

Therefore, these questions need to be revised to be published. I recommend a major revision.

Reviewer 3 Report

Type manuscript: Article – original research manuscript.

Title manuscript: Environmental impact of the average Hong Kong diet: A case for adopting sustainable diets in urban centers

Short characteristic manuscript:

In the manuscript ‘Environmental impact of the average Hong Kong diet: A case for adopting sustainable diets in urban centers’, the Authors raise very important an issue about sustainable diets, and their impact to environmental. It’s a big problem in the world. The Authors tried to estimate the carbon and water footprints of the Hong Kong average diet from available sources, and compare to well accepted sustainable diets. Additionally, the Authors tried to characterize environmental sustainability.

General comments for Authors:

Generally the manuscript provides valuable information. However, I have some remarks.

The first general concern is that at the end of the Introduction section, the exact purpose of the study is missing. Thus, the introduction needs to be re-written with that in mind.

Additionally, Methods and Materials section should be after Introduction section.

There should not be such large intervals between paragraphs.

You should unify punctuation marks. Once, there is a long dash, and once short.

Moreover, bibliography needs proofreading.

Comments and detailed suggestions for Authors:

Line 87;

There should be no space between 50 and %.

Table 3;

This table is unnecessary. Bibliography is given at the end.

Line 232;

Names of food groups should be in lowercase.

Line 250-255;

This paragraph should be at the end of the Introduction section.

Author Response

Reviewer 3:

Type manuscript: Article – original research manuscript.

Title manuscript: Environmental impact of the average Hong Kong diet: A case for adopting sustainable diets in urban centers

Short characteristic manuscript:

In the manuscript ‘Environmental impact of the average Hong Kong diet: A case for adopting sustainable diets in urban centers’, the Authors raise very important an issue about sustainable diets, and their impact to environmental. It’s a big problem in the world. The Authors tried to estimate the carbon and water footprints of the Hong Kong average diet from available sources, and compare to well accepted sustainable diets. Additionally, the Authors tried to characterize environmental sustainability.

General comments for Authors:

Generally the manuscript provides valuable information. However, I have some remarks.

The first general concern is that at the end of the Introduction section, the exact purpose of the study is missing. Thus, the introduction needs to be re-written with that in mind.

Response: Thank you for your suggestion. The end of the introduction has been reworded and expanded to further evaluate our purpose in doing this study. “In the case of Hong Kong, the most recent reports on GHG emissions only measure water consumption [26] and the average diet [7]. Thus, the need to systematically analyze and report the environmental impacts of a typical Asian or Hong Kong local dietary habit using multiple indicators is warranted. By doing so, this study hopes to highlight the problems associated with current consumers’ dietary habits and advocate for a change that will positively impact the environment.

The objective of this study is to characterize the environmental sustainability of a mean Hong Kong diet by comparing its calculated impact to three well-accepted sustainable diets: 1) the vegan diet, 2) the Mediterranean diet, and 3) the New Nordic diet [27, 28]. To our knowledge, no comparable studies on the WF for sustainable diets or a recommended WF value for a food system are currently available. The estimated CF of the Hong Kong diet was also compared to a recommended sustainable value of 2055g CO2-eq per capita per day [21].” Lines 96-107, Introduction.

Additionally, Methods and Materials section should be after Introduction section.

Response: Thank you for your comment but according to Instructions to Authors on the journal’s website and the journal’s template, Materials and Methods is placed after Discussion and before Conclusion. We will clarify with the editorial team and are happy to move this section as suggest by the reviewer.

There should not be such large intervals between paragraphs.

Response: Thank you for your comment. Intervals have been eliminated and paragraphs are closer to each other now.

You should unify punctuation marks. Once, there is a long dash, and once short.

Response: Thank you for your comment. All punctuation marks have been unified, and the long dash has been updated

Moreover, bibliography needs proofreading.

Response: Thank you for your comment. The entire bibliography has been proofread and updated.

Comments and detailed suggestions for Authors:

Line 87; There should be no space between 50 and %.

Response: Thank you for your comment. The space has been removed.

Table 3; This table is unnecessary. Bibliography is given at the end.

Response: Thank you for your comment. Table 3 has been deleted.

Line 232; Names of food groups should be in lowercase.

Response: Thank you for your comment. Names of the food groups have been changed to lowercase.

Line 250-255; This paragraph should be at the end of the Introduction section

Response: Thank you for your suggestion. This paragraph has been reworded and moved to the end of the introduction section. “The objective of this study is to characterize the environmental sustainability of a mean Hong Kong diet by comparing its calculated impact to three well-accepted sustainable diets: 1) the vegan diet, 2) the Mediterranean diet, and 3) the New Nordic diet [27, 28]. To our knowledge, no comparable studies on the WF for sustainable diets or a recommended WF value for a food system are currently available. The estimated CF of the Hong Kong diet was also compared to a recommended sustainable value of 2055g CO2-eq per capita per day [21].” Lines 102-107, Introduction.

Round 2

Reviewer 2 Report

The authors have improved the manuscript and the comments raised have been discussed and responded satisfactorily. This has made me understand that the main objective of this article is environmental. I hope that in the future this study can be completed with a nutritional impact study.